# Ldha-Dependent Metabolic Programs in Sertoli Cells Regulate Spermiogenesis in Mouse Testis

**DOI:** 10.3390/biology11121791

**Published:** 2022-12-09

**Authors:** Xiao-Na Zhang, Hai-Ping Tao, Shuang Li, Yu-Jun Wang, Shi-Xin Wu, Bo Pan, Qi-En Yang

**Affiliations:** 1Key Laboratory of Adaptation and Evolution of Plateau Biota, Northwest Institute of Plateau Biology, Chinese Academy of Sciences, Xining 810008, China; 2University of Chinese Academy of Sciences, Beijing 100049, China; 3Farm Animal Genetic Resources Exploration and Innovation Key Laboratory of Sichuan Province, College of Animal Science and Technology, Sichuan Agricultural University, Chengdu 611130, China; 4Qinghai Key Laboratory of Animal Ecological Genomics, Northwest Institute of Plateau Biology, Chinese Academy of Sciences, Xining 810001, China

**Keywords:** Sertoli cells, lactate, metabolism, spermiogenesis, fertility, spermatogenesis, choline

## Abstract

**Simple Summary:**

Infertility affects an estimated 10–15% of couples, and males are responsible for 20–30% of infertility cases worldwide. Male fertility relies on the success of spermatogenesis, which is a complex cellular differentiation process that is coordinated by germ cells and testicular somatic cells. Sertoli cells play a central role in supporting spermatogenesis, and it was proposed that they supply energy substances for the developing germ cells. In the present study, we provide genetic evidence that lactate production in Sertoli cells, which is controlled by the lactate dehydrogenase A (*Ldha*) gene, is crucial for murine spermatogenesis. Compared to the controls, *Ldha* deficiency in Sertoli cells greatly reduced the testicular weight, which caused spermatogenic defects and adversely impacted sperm function. The sperm from *Ldha* conditional knockout animals exhibited low motility, high morphological abnormalities, and an impaired capacity to fertilize oocytes. Metabolic analysis has revealed that limiting lactate production in Sertoli cells changed a large number of metabolites in the sperm from mutant animals. We then identified choline as the key molecule that mediates the function of lactate in sperm, and the supplementation of conditional knockout animals with dietary choline rescued the spermatogenic defects and fertility. This work sheds new light on the functional role of Sertoli cells in regulating spermatogenesis and identifies a candidate molecule that has potential roles in treating male infertility that is caused by metabolic disorders.

**Abstract:**

Sertoli cells play indispensable roles in spermatogenesis by providing the advanced germ cells with structural, nutritional, and regulatory support. Lactate is regarded as an essential Sertoli-cell-derived energy metabolite that nurses various types of spermatogenic cells; however, this assumption has not been tested using genetic approaches. Here, we have reported that the depletion of lactate production in Sertoli cells by conditionally deleting lactate dehydrogenase A (*Ldha*) greatly affected spermatogenesis. *Ldha* deletion in Sertoli cells significantly reduced the lactate production and resulted in severe defects in spermatogenesis. Spermatogonia and spermatocytes did not show even mild impairments, but the spermiogenesis of *Ldha* conditional knockout males was severely disrupted. Further analysis revealed that 2456 metabolites were altered in the sperm of the knockout animals, and specifically, lipid metabolism was dysregulated, including choline, oleic acid, and myristic acid. Surprisingly, choline supplementation completely rescued the spermiogenesis disorder that was caused by the loss of *Ldha* activities. Collectively, these data have demonstrated that the interruption of Sertoli-cell-derived lactate impacted sperm development through a choline-mediated mechanism. The outcomes of these findings have revealed a novel function of lactate in spermatogenesis and have therapeutic applications in treating human infertility.

## 1. Introduction

Spermatogenesis occurs in the seminiferous tubules of the testes, and this complex process is supported by testicular somatic cells. Sertoli cells are the only somatic cells in the seminiferous tubules that directly contact the spermatogenic cells in order to regulate spermatogonial proliferation, meiosis, and postmeiotic spermatid development [1]. The Sertoli cells produce niche factors in order to support the establishment and the maintenance of spermatogonial stem cells (SSCs) [2], generate retinoic acid to direct spermatogonial differentiation and meiosis initiation [3,4], and control the terminal differentiation of spermatids through androgen signaling [5]. Because of their crucial functions in providing physical, nutritional, and regulatory support, the Sertoli cell number ascertains the testis property and the sperm output by regulating the germ cell and Leydig cell populations [6]. Defects in Sertoli cell development and function consequentially impact fertility and reproductive health.

The Sertoli cells are major sources of energy substances for the developing germ cells that are separated from the other testicular somatic cells by the basement membrane. Sertoli cells are large and irregularly shaped, which creates a unique microenvironment for meiotic and postmeiotic germ cell development and maturation in the seminiferous tubules [7]. Sertoli cells secrete large amounts of fluid that contains hormones, transferrin, growth factors, lactate, and other substrates [8]. The developing spermatogenic cells utilize lactate as their major energy source [9]. Sertoli cells obtain glucose and generate lactate through glycolysis, which is catalyzed by different enzymes in a series of 10 reactions [10]. The final process of glycolysis involves the reversible conversion between pyruvate and lactate, which relies on the activities of lactate dehydrogenase (LDH) [11]. LDH is a tetrameric glycolytic enzyme consisting of four different subunits that function as a cohesive unit [12]. At least four genes encode this enzyme in animals; specifically, LDHA, LDHB, and LDHC encode the L-isomers, while LDHD encodes the D-isomer [12,13]. L-Lactate is the predominant form of lactate in animals, and it has been proposed to be supplied by the Sertoli cells in the testis [14]. Spermatocytes and mature sperm prefer to use lactate as fuel in order to generate ATP [15]. The survival of cultured spermatocytes and spermatids is regulated by lactate supplementation, not glucose, in vitro [16,17]. These experiments provide important data that indicate an essential role of Sertoli-cell-derived lactate in spermatogenesis. However, genetic evidence that demonstrates the specific requirement of lactate from the Sertoli cells for spermatogenesis is lacking.

All of the LDH gene family members are expressed in the testis, and *Ldhc* has crucial functions in spermatogenesis. *Ldha* and *Ldhb* are ubiquitously expressed genes that are also detected in the Sertoli cells of mouse testis [18]; in contrast, *Ldhc* is only present in postmeiotic germ cells, and its expression is required for spermatogenesis [19]. The sperm of *Ldhc*-null mice exhibits a variety of impairments, including decreased ATP levels and defects in their motility and capacitation [19,20]. The sterile phenotype of *Ldhc* knockout mice can be rescued by the human LDHA transgene; however, decreased ATP and lactate levels remain in transgenic animals, indicating the functional similarity and specificity of these two genes [21]. In addition, lactate serves a dual role as an energy supply substrate and as a multifunctional signaling molecule that directs a variety of cellular processes [22,23]. Lactate is a ligand for hydroxycarboxylic acid receptor (Hcar-1), and the activation of lactate-Hcar1 signaling directs neuron activities, intestinal homeostasis, and other key cellular events [24,25]. Conditional knockout models have been widely used in order to dissect the genetic regulation of Sertoli cell function [26], and specific deletion of *Ldha* or *Ldhb* is crucial for deciphering the actions of lactate and its related molecular mechanisms in spermatogenesis. However, a detailed examination of LDH function in Sertoli cells has not been reported in vivo.

Several lines of evidences suggest that LDHA exhibits a higher affinity for pyruvate and it is the major enzyme that is responsible for the conversion of pyruvate into lactate [27,28]. *Ldha* is indicated to be the critical regulator of lactate synthesis in Sertoli cells [29], and its expression is tightly controlled by follicle-stimulating hormone (FSH) [30]. In this study, we selectively deleted *Ldha* expression using the Cre-LoxP system in Sertoli cells in order to decipher the functional roles of *Ldha*-dependent lactate in murine spermatogenesis. The results showed that *Ldha* loss of function in Sertoli cells caused a significant decline in fertility, impaired spermatogenesis, and changed the metabolic profile of sperm.

## 2. Materials and Methods

### 2.1. Animals

All of the animal studies were performed based on the guidelines of the Institutional Animal Care and Use of Laboratory Animals and were approved by the Animal Welfare and Ethics Committee at the Northwest Institute, Chinese Academy of Sciences. Sox9-eGFP mice were obtained from Mutant Mouse Resource and Research Centers (MMRRC_011019-UCD). *Ldha^fl/fl^* mice (The Jackson Laboratory, 030112, Bar Harbor, USA) were crossed with *Stra8-cre* mice to obtain *Stra8-cre*; *Ldha^fl/+^ mice*, which were mated with *Ldha^fl/fl^* mice to generate Ldha conditional knockout mice (*Stra8-cre*; *Ldha^fl/f^*). *Ldha^fl/fl^* mice were crossed with *Amh-cre* mice to obtain *Amh-cre*; *Ldhaf^l/+^* mice, which were mated with *Ldha^fl/fl^* mice to generate *Ldha* conditional knockout mice (*Amh-cre*; *Ldha^fl/fl^*, referred to hereafter as *Ldha*-cKO animals). The *Amh-cre*; *Ldha^fl/+^* littermates were used as controls. *Amh-cre* and *Ldha* LoxP sites were identified using genotyping polymerase chain reaction (PCR). The product sizes and primers are listed in Appendix A. For each experiment, at least three different mice (*n* > 3 biological replicates) were used for each genotype. For the fertility test, adult control and *Ldha*-cKO males (*n* > 3 for each genotype) were paired with wild-type females. Each male (2 months old) was paired with 4 females for 3 months, and the litter sizes were quantified.

### 2.2. Histological, Immunofluorescent, and Apoptosis Analyses

Histological analysis was performed as described previously [31]. Briefly, the testes were fixed in Bouin’s solution and embedded in paraffin solution. Sections were cut to a thickness of 5 μm, and testicular cross-sections were stained with hematoxylin and eosin (H&E) and Periodic Acid–Schiff (PAS) staining according to standard protocols [32]. For immunofluorescent staining, the testes were fixed in 4% paraformaldehyde solution (PFA), and the slides were boiled in 10 mM sodium citrate buffer for antigen retrieval. Nonspecific binding was blocked with 10% goat serum albumin at room temperature (RT) for 1 h and then incubated with a primary antibody at 4 °C overnight. The slides were then incubated with the appropriate secondary antibody for 2 h. Terminal deoxynucleotidyl transferase (TdT)-mediated dUTP-biotin nicked end-labeling (TUNEL), which is always used to detect apoptosis, was performed by using an In Situ Cell Death Detection kit (TUNEL, Beyotime, Shanghai, China) to detect DNA fragmentation in the apoptotic cells. DNA was visualized using H33342 staining, and all images were acquired with a microscope (Nikon ECLIPSEE200, Tokyo, Japan) that equipped with a Hamamatsu CCD camera (MshOt MS60, Guangzhou, China). The antibodies used in the study are listed in Appendix A.

### 2.3. Sertoli Cell Isolation and Quantitative RT-PCR

Sertoli cells were isolated from *Sox9-eGFP* transgenic mice using fluorescence-activated cell sorting, as described previously [33]. Total cellular RNA was extracted using TRIzol reagent (Invitrogen, Carlsbad, CA, USA). The quality and concentration of RNA were measured by using a Nanodrop 2000c Spectrophotometer (Thermo Fisher, Waltham, MA, USA). The cDNA synthesis was conducted by using the High-Capacity cDNA Reverse Transcription kit (Invitrogen, Carlsbad, CA, USA). Specific primers were used to quantify the relative abundance of transcripts using a ViiA7 Real-Time PCR System (Applied Biosystems, Foster, CA, USA). The conditions for the thermal cycles were as follows: activation at 95 °C for 2 min, accompanied by 40 cycles of 95 °C for 20 s and 60 °C at 30 s each. *Gaphd* was used as an internal reference, and three replicates were conducted using different biological samples. The relative RNA expression was computed by the 2^−∆∆ct^ method, as described previously [34].

### 2.4. Sperm Quality and Ultrastructural Analyses(TEM)

Epididymal sperm were collected in 300 μL human tubal fluid (HTF) and were processed for PNA and mitochondrial staining with peanut agglutinin (PNA), as described previously [35]. The testes were fixed in 4% paraformaldehyde solution (PFA), and the slides were boiled in 10 mM sodium citrate buffer for antigen retrieval. Then, the PNA (1:500, L32458, Invitrogen™) with red fluorescence was incubated at 4 °C overnight. Then, DAPI was added. Computer-assisted sperm analysis (CASA) was used to determine the sperm quality (WLJY-9000, WEI-LI New Century Technical Development, Beijing, China), including (i) percentage of sperm viability, (ii) percentage of sperm motility, (iii) sperm density, and (iv) sperm velocity (average path velocity, curvilinear velocity, and straight-line velocity). To examine ultrastructural changes due to *Ldha* deletion in Sertoli cells, sperm were obtained from the caudal epididymis of the control and *Ldha*-cKO mice and fixed in 2.5% glutaraldehyde at 4 °C overnight, as described previously [36]. Then, the material was dehydrated and embedded in Epon resin; ultrathin sections were immersed with 3% uranyl acetate and lead citrate and analyzed with a JEOL JM-1400Plus electron microscope (Tokyo, Japan), which was used to capture images at 6000× and 12,000× magnifications.

### 2.5. Unlabeled Targeted Metabolomics

To investigate the changes in sperm metabolism caused by *Ldha* loss of function, sperm samples (n = 6 for control and n = 7 for *Ldha*-cKO animals) were collected and lysed in 80% methanol at 4 °C. LC-MS/MS analysis was carried out on the metabolite extracts using Applied Protein Technology, Shanghai, China (Applied Protein Technology). We used the LC-MRM methods to measure the retention times for the listed compounds, and we created final LC-dynamic MRM methods based on delta retention time windows of 1.4 to 3.4 min. Agilent Mass Hunter Quantitative Analysis (for QQQ) Version B.07 was used to process the initial data. A software analysis was used to integrate the peak areas of detected compounds, and a manual inspection was conducted to verify their accuracy. Normalizing metabolite counts to reflect equal loading of all samples was then performed by dividing all metabolite counts by their total intensities. For statistical analyses and visualization, each metabolite abundance level was divided by the median of all abundance levels across all of the samples. We performed Student’s *t* test with a significance level of 0.05 using a two-tailed Student’s *t* test. We did not adjust the *p* values to accommodate for a more flexible biological interpretation and downstream visual inspection of the data. In Metabo Analyst 4.0, an enrichment analysis was performed on metabolic pathway enrichment [37]. The software is available at https://www.metaboanalyst.ca (MetaboAnalyst5.0, accessed on 5 November 2020).

### 2.6. Lactate Concertation Measurement

The lactate concentration of Sertoli cells was determined with a lactic acid assay kit (H700059, Completed in Nanjing). The supernatants were collected after centrifugation for 10 min at 2000 rpm. Then, the samples were diluted and processed according to the manufacturer’s protocols. Finally, a terminating agent was added to terminate the reaction, and the absorbance at 530 nm was recorded using a microtiter plate reader. The OD value of the blank control well was set as the background.

### 2.7. Choline Supplementation

Unless stated otherwise, *Ldha*-cKO mice were given choline (CAS: 67-48-1, Sigma, Darmstadt, Germany) diluted in PBS at 2.5 mg/mL (20 mM) at day 20 after birth for 25 days in their drinking water, while the nontreated *Ldha*-cKO groups received drinking water supplemented with the same volume of PBS that was used to dilute choline at 20 mM. The drinking water supply was replenished twice per week. A total of 1.1 g/kg choline daily supplementation at this dose is well tolerated in mice [38]. For the fertility test, the choline-treated *Ldha*-cKO mice and PBS-treated males (n = 3 for *Ldha*-cKO with added PBS and n = 5 for *Ldha*-cKO receiving choline treatment) were paired with wild-type females. Each male was paired with 4 females for 3 months, subsequent the litter sizes were quantified. Then, the testes were collected for H&E and PASstaining.

### 2.8. Sperm Membrane Staining and Acrosomal Reaction (AR) Analysis

Sperm with intact cell membranes were identified by a dye exclusion test, and the percentage of viable spermatozoa was determined by eosin-aniline black staining (Leagene, Beijing, China). The percentage of viable sperm was determined by dye rejection to identify the sperm with intact cell membranes. That is the damaged cell membrane allows non-permeable dyes to enter the membrane and stain it, while the cell membrane of living cells can resist the entry of dyes, resulting in the phenomenon of dye rejection and no staining. The sperm–zona pellucida (ZP) binding assay was performed as described previously [39]. Briefly, cumulus-free oocytes were placed in a drop of human tubal fluid (HTF) medium and inseminated with capacitated sperm (2 × 10^5^ sperm/mL). After incubation at 37 °C in 5% CO_2_ in air for 30 min, the oocytes were fixed with 0.25% glutaraldehyde, and the number of sperm bound to the ZP was counted after DAPI staining. At least 50 oocytes were analyzed for each group. To evaluate the acrosomal reaction (AR), the mature spermatozoa isolated from the cauda epididymis were incubated in HTF medium at 37 °C, 5% CO_2_ (*v*/*v*), and 95% (*v*/*v*) air for 2 h. The capacitated spermatozoa were treated with calcium ionophore A23187 (C9400-5MG; Sigma, Germany) to induce AR. The acrosome was stained with fluorescein-conjugated (PNA) lectin labeling (50 mg/mL), and the sperm nuclei were stained with DAPI (5 mg/mL). Two hundred sperm were used to analyze the process of anomaly and acrosomal reaction detection. The percentage of spermatozoa without acrosomes was assessed under a fluorescence microscope and taken to be the percentage of acrosome-reacted spermatozoa (AR ratio).

### 2.9. In Vitro Fertilization and Embryo Culture

Spermatozoa were incubated in HTF medium for 1.5 h, and oocytes were collected using superovulation, as described previously [39]. The female mice were injected with 5 IU pregnant mare serum gonadotropin (PMSG), followed by 72 h injection of 5 IU human chorionic gonadotropin (HCG). Then, the bulge of the fallopian tube was harvested in M2 solution at 37 °C. Then, the oocytes were incubated with sperm. At 4 h postfertilization, the oocytes were removed from the fertilization drop, washed in KSOM medium (Millipore), transferred to drops of KSOM medium covered with mineral oil, and then cultured in an incubator for 24 h and 96 h to observe the 2-cell and blastocyst rates, respectively.

### 2.10. Western Blot

Western blotting was performed as previously described [31]. The testes from the *Lhda*-cKO mice and the controls were harveste. Then, the enhanced cracking fluid (17C11B02, BOSTER, Wuhan, China) was used to lyse the testis, which was further centrifuged at 12,000× *g* at 4 °C for 30 min. Equal amounts of protein (45 μg) were loaded onto a 5–15% SDS-PAGE gel (Cat#P1200, Solarbio, Beijing, China), electrophoresed, and transferred to cellulose acetate membranes (0.45 µm), which were blocked with 5% nonfat blocking-grade milk. Primary antibodies (LDHA, 1:1000, AB101562, Cambridge UK; β-actin, 1:2000, GB11001, Servicebio, Wuhan, China) were incubated with the membranes overnight at 4 °C. The secondary antibody was used properly at room temperature for 2 h (goat to rabbit, 1:4000, BA1032, Boster, China; goat to mouse, 1:4000, BA1031, Boster, China). ECL Plus chemiluminescence reagent kit (AR1197, BOSTER, China) was used captured the bands, which were quantified with optical methods using ImageJ software (4200SF, Tanon, Shanghai, China). The results were normalized using β-actin as a control.

### 2.11. Statistical Analysis

Data are presented as the means ± s.e.m.s for at least 3 independent experiments. At least 3 animals were used for each genotype. Sertoli cells were identified by SOX9 staining. Germ cells were identified by LIN28A or SYCP3 staining. In the analysis of the Sertoli cell numbers per cord, 30 tubules were used on average per mouse. An average of 1000 Sertoli cells per mouse were used for germ cell analysis. The number of Sertoli cells used to analyze the percentage of TUNEL+ Sertoli cells was 2000 for each genotype. A *t* test was used to compare the differences between the two groups. One-way ANOVA was used to compare the difference when more than 2 groups were compared. The differences between means were examined using the *t* test function of GraphPad Prism 7 (La Jolla, San Diego, CA, USA), and were considered significant at *p* < 0.05.

## 3. Results

### 3.1. Ldha Expression Was Enriched in Sertoli Cells, and Conditional Deletion of Lhda Significantly Impacted Fertility

*Ldha* is the predominant form of LDH transcript in mouse Sertoli cells, and its expression is tightly regulated by hormones and transcription factors that have important roles in spermatogenesis [29,40]. First, we isolated *Sox9^+^* cells from wild-type adult testes and quantified the relative abundances of *Ldha*, *Ldhb*, *Ldhc,* and *Ldhd* mRNA. *Ldha* and *Ldhb* transcripts were detected in the Sertoli cells, and the relative expression of *Ldha* was 47-fold higher than that of *Ldhb* (*p* < 0.05) (Figure 1A). We then examined the relative abundance of *Ldha* at different developmental stages, including postnatal days (PDs) 0, 6, 14, 21, and 35. During neonatal development, the *Ldha* transcript was expressed at low levels in the testes, but increased by 16.59 ± 0.79-fold at PD21 (*p* < 0.05) (Figure 1B). Immunofluorescent co-staining for LDHA and GATA4 revealed that LDHA was present both in the neonatal and the mature Sertoli cells (Figure 1C). These data supported the previous data that LDHA is abundantly present in Sertoli cells, indicating their potential roles in spermatogenesis. *Ldha* expression was also detected in the spermatocytes and the spermatids, and we generated a conditional knockout mouse model in which *Ldha* was deleted in the spermatogenic cells using the Cre-LoxP methodology. The results showed that the deletion of *Ldha* using *Stra8-cre* in the spermatogenic cells did not affect the spermatogenesis or the fertility (Appendix A). Therefore, we focused on LDHA function in Sertoli cells in the present study.

We detected whether *Ldha* in Sertoli cells was crucial for spermatogenesis and fertility. To this end, we generated a mouse line that was lacking *Ldha* in the Sertoli cells using *Amh-cre*-mediated conditional deletion. *Amh-cre*-induced recombination occurs around day 14.5 of embryonic development; therefore, *Ldha* was genetically eliminated in the Sertoli cells of fetal origin. The Western blot results showed that *Ldha* was reduced in the Sertoli cells (Figure 2A). The quantitative density analysis of proteins revealed that LDHA expression was greatly reduced (1.08 ± 0.03-fold), but not completely absent, in the testes of the *Ldha*-cKO animals compared to the controls (0.27 ± 0.05), as *Ldha* was also expressed in the germ cells (Figure 2B). The quantitative analysis revealed that *Ldha* expression was reduced by 56% in the testes of the *Ldha*-cKO animals (0.98 ± 0.08 vs. 0.44 ± 0.35) (Figure 2C). As expected, the content of lactic acid that was produced by the Sertoli cells was decreased by more than 90% in the *Ldha*-cKO testes compared to that in the control testes (Figure 2D). The testis weight of the *Ldha*-cKO animals was reduced by 44.26% compared to the controls, and the epididymis weight of the *Ldha*-cKO animals was also significantly reduced by 24.21% at three months of age (Figure 2E,F and Appendix A).

The fertility test revealed that *Ldha* deletion severely impacted fertility because only 3 of the 15 *Ldha*-cKO mice sired pups (10.41 ± 0.49 vs. 0.60 ± 0.34) and the litter size was significantly reduced (Figure 2G). A detailed examination of the sperm quality and quantity showed that the sperm concentration in the *Ldha*-cKO mice was reduced by 99.46% compared to the controls (3.6 × 10^5^ ± 0.70/mL vs. 6.7 × 10^7^ ± 1.6/mL control) (Appendix A). The rate of sperm abnormality was increased by 247%, and the viability was decreased by 41.92%. The parameters of sperm motility were all decreased significantly in the *Ldha*-cKO mice (Appendix A). The sperm were classified into four grades according to the World Health Organization, A, B, C, and D, according to their motility, where grade A indicated rapid forward movement, in which the velocity should be ≥25 μm/s under the microscope, and 25 μm is five times the length of a sperm head, or half the length of a sperm tail. Grade B indicated a slow forward movement, in which the velocity should be ≤25 μm/s, grade C indicated in situ or a curved movement, but sperm could still be active, and grade D indicated immobility, and some of the sperm may be live and some may be dead. As a result, the percentages of the grade A and grade B sperm were decreased by 34.12% and 46.10%, respectively, while the percentage of the grade D sperm was increased by 64.63% (Appendix A). Additionally, apoptosis in the testis was increased by 132.68% in the *Ldha*-cKO mice compared to the controls (Appendix A). Collectively, these data supported the conclusion that *Ldha* deletion severely impacted the sperm motility and quality, thus causing a decline in fertility.

### 3.2. Ldha Loss of Function in Sertoli Cells Resulted in Defects in Spermiogenesis

In order to investigate the cause of the significantly decreased fertility in the *Ldha*-cKO males, we performed a histological analysis on the adult testes. The seminiferous tubules of the control males contained three layers of spermatogenic cells with normal spermatogenesis; in sharp contrast, the spermatogenesis was impaired in the testes of the *Ldha*-cKO males, and the most obvious phenotype was defects in spermatid differentiation (Figure 3A). The number of spermatogonia and spermatocytes per round tubule was slightly decreased between the control and the *Ldha*-cKO mice (Appendix A). The number of round spermatids was decreased by 27.18% (Figure 3C). Notably, abnormal developmental spermatid was evident, and the stages of seminiferous tubules were disorganized (Figure 3A,B). Testicular cross-sections from the testes of the control and the *Ldha*-cKO adult males were stained with PASin order to evaluate the spermatogenic stages in detail. Spermatogenesis can be divided into 16 steps in mice, which also define the seminiferous tubules into stages I-XII, along with the cellular association between the developing germ cells and the Sertoli cells [41]. The results showed that the percentage of seminiferous tubules at stages I-III, VII-VIII, and IX-XII was significantly decreased, while that at stages IV-VII it was increased in the testes of the *Ldha*-cKO animals compared to the controls (Figure 3A,B), indicating that spermiogenesis was likely blocked at this stage. PNA labeling, which recognizes the acrosomal status, revealed that the number of spermatids at steps 7–8 was decreased in the seminiferous tubules of the *Ldha*-cKO testes (Appendix A). Together, these findings indicated that *Ldha* deletion in the Sertoli cells impacted the process of spermiogenesis.

### 3.3. Ldha Deletion in Sperm Affected Plasma Membrane Integrity and Function

In order to further examine the effects of the Sertoli-cell-specific *Ldha* knockout on sperm function, we first examined the plasma membrane integrity, which is vital for sperm function [42]. The transmission electron microscopy (TEM) analysis of the sperm revealed ultrastructural abnormalities that were caused by *Ldha* loss of function in the Sertoli cells. The plasma membrane of the control sperm was intact; however, it was severely damaged in the *Ldha*-cKO animals (Figure 4A). Eosin-aniline black staining validated these findings and revealed that 72.67% of the sperm contained damaged plasma membranes (Figure 4C,D). Impaired microtubule structure and mitochondrial defects were observed in a fraction of the sperm from the *Ldha*-cKO animals (Figure 4B,E).

Due to the abovementioned changes, the acrosome reactions that were induced by A23187 were completely different in the sperm from the *Ldha*-cKO animals compared to the control mice. (Figure 4F). The percentage of sperm that was attached to the oocytes was significantly affected by *Ldha* deletion in the Sertoli cells, indicating that the fertilization capacity of the sperm that was produced by the *Ldha*-cKO mice was defective (Figure 4G,H). Indeed, the in vitro fertilization (IVF) experiments revealed that the 2-cell and blastocyst rates of sperm from the *Ldha*-cKO males were decreased by more than 5.76% and 40.71%, respectively, compared to the controls (Appendix A). Collectively, these results provide strong evidence that LDHA function in Sertoli cells is crucial for maintaining the plasma membrane integrity and the fertilization ability of sperm in mice.

### 3.4. Ldha Deletion in Sertoli Cells Changed the Metabolic Profile of Sperm

Next, we conducted metabolomics analysis in order to screen the candidate molecules that mediate the actions of the Sertoli-cell-derived lactate in sperm. To this end, untargeted metabolomics analysis were performed on the epididymal sperm from the control (n = 6) and the *Ldha*-cKO animals (n = 7). After stringent quality control (Appendix A), a total of 18881 metabolites were detected in the control sperm, and 16788 were identified in the sperm from the *Ldha*-cKO animals (Appendix A). A total of 480 metabolites were identified in the database; among them, 47 metabolites were upregulated (Appendix A). and 433 were downregulated (Appendix A) in the sperm of the *Ldha*-cKO animals (Figure 5A, Appendix A, Appendix A). Further analysis identified that the metabolism of linoleic acid, unsaturated fatty acids, glycerophospholipids, galactose, and glycerolipids were upregulated(Appendix A), while the citrate cycle, glycolysis, phenylalanine metabolism, and pyruvate metabolism were downregulated (Figure 5B, Appendix A). Myistic acid, linoleic acid, phosphocreatine, pentadecanoic acid, and N-acetyl-D-Glucosamine 6-phosphate were significantly increased in the sperm of the *Ldha*-cKO mice compared to the control animals. In contrast, N6-methyladenosine, uracil, prostaglandin D2 (PGD2), citrate, alpha-D-galactose 1-phosphate and orotate, acetylcarnitine, glycerol 3-phosphoethanolamine, and L-phenylalanine were decreased in the sperm of the *Ldha*-cKO mice compared to the controls (Figure 5C, Appendix A). This indicated that the decreased fertilization capacity of the sperm from the *Ldha*-cKO animals may be caused by the increase in lipid metabolism and the decline in glycolic metabolism. Through the captured peak, it was observed that oleic acid, N6-methyladenosine, citrate, uracil and choline concentrations were decreased (Appendix A). We particularly focused on the downregulated metabolites that have a potential role in spermatogenesis and found that choline was the top candidate in this context (Figure 5C). Choline is the main component of the cell membrane, and choline phospholipids, phosphatidylcholine plasmalogen, and sphingomyelin (SPM) are present on the sperm plasma membrane [43]. Normal fertility requires an adequate amount of choline, and genetic defects in choline biogenesis are associated with abnormal sperm function in both humans and mice [44,45]. The correlation analysis of the differential metabolites indicated that choline was associated with glycerophosphocholine, phosphorylcholine, and other choline-containing metabolites (Appendix A).

### 3.5. Choline Supplementation Rescued Spermatogenic Defects in Ldha-cKO Mice

Due to the lactate reduction in the Sertoli cells, we isolated the sperm in vitro cultivated with 20 mM lactate in HTF in the *Ldha*-cKO mice, while the *Ldha*-cKO mice with added dimethyl sulfoxide were as the controls. The result showed that the abnormal sperm treated with 20 mM lactate was not different compared to the controls (108.81 ± 4.35 vs. 117.61 ± 4.10) (Appendix A). The membrane of the sperm was also tested, and there was no difference in the sperm that was treated with 20 mM lactate compared to the controls (146.80 ± 3.61 vs. 144.01 ± 3.67) (Appendix A). As expected, the number of sperm that adhered to the oocytes was also not different between the 20 mM lactate-treated sperm and the control sperm (5.73 ± 0.72 vs. 6.13 ± 0.66) (Appendix A). This may be due to the sperm membrane damage in the *Ldha*-cKO mice (Figure 4A). We reasoned that the decreased choline content in the sperm likely resulted in the impairment of the membrane integrity and the sperm motility. We supplemented the controls (n = 3) and the *Ldha*-cKOmice (n = 5) with choline in their drinking water for 25 days and examined the fertility and the spermatogenesis. The results showed that choline supplementation effectively rescued the fertility, indicating that exogenous choline at this concentration did not cause adverse effects on spermatogenesis. In sharp contrast, the testis/body weight ratio of the *Ldha*-cKO animals receiving the choline treatment increased significantly compared to the controls that deal ed with same volume of PBS (Figure 6A,B). The pregnancy rate in the *Ldha*-cKO mice treated with PBS was 6.67%, while in the *Ldha*-cKO mice treated with choline was 83.33%,, which indicated an observable rescue of the reproductive efficiency by the choline treatment (Figure 6C). The fertility test showed that the litter size of the choline-treated *Ldha*-cKO animals was 8.0 ± 0.84, which was similar to that of the wild-type controls, indicating a complete rescue of the fertility by the choline treatment (1.42 ± 0.68) (Figure 6D). The histological analysis confirmed that the spermatogenic defects, including disorganized spermiogenesis, disappeared and the sperm concentration increased dramatically (Figure 6E,F). As expected, the abnormal sperm in the choline-treated *Ldha*-cKO mice were decreased compared to those in the PBS-treated mice (64.37 ± 3.15 vs. 11.30 ± 0.60) (Figure 6G,H). Importantly, an examination of the sperm defects revealed that the percentage of abnormal sperm was comparable between the control and the *Ldha*-cKO mice after choline supplementation. Collectively, these data provided evidence that the defects in the sperm morphology and function that were caused by *Ldha* deletion in the Sertoli cells were rescued by choline supplementation.

## 4. Discussion

The findings of the present study provide important evidence that *Ldha* in Sertoli cells serves crucial functions in maintaining normal fertility by directing spermiogenesis. *Ldha* deletion in Sertoli cells caused a significant decline in the sperm number and caused severe defects in the sperm motility, the membrane integrity, and the fertilization capacity. The metabolic profiling of sperm from the mutant animals identified choline as the key molecule that mediates the function of lactate in sperm. Overall, the present study established a crucial role of *Ldha*-dependent lactate in spermatogenesis and uncovered a novel functional link between lactate and choline. The outcomes of these findings have potential clinical implications in lactate-related human infertility or other diseases.

*Ldha* is likely the predominant form of LDH in Sertoli cells, and its expression is tightly regulated. The protein that is encoded by the *Ldha* gene exists as a tetramer and is known as the M subunit. Its main function is to catalyze pyruvate to lactate and convert nicotinamide adenine dinucleotide (NAD)H to NAD+ [28]. At the transcriptional level, the promoter region of the *Ldha* gene contains binding motifs for hypoxia inducible factor 1 (*Hif1*), c-Myc, Kruppel-like factor 4 (*Klf4*), and forehead box protein M1 (*Foxm1*) [40,46,47,48]. FSH likely works in concert with these and other unknown factors in order to control *Ldha* transcription in the Sertoli cells [48]. The relative abundance of *Ldha* mRNA was low in testes from the 6- to 12-day-old mice, but gradually increased in the testes from postnatal PD16 to PD45, and its expression was regulated by differential methylation [49]. In this study, we found that the relative abundance of *Ldha* was significantly higher than that of *Ldhb* in the Sertoli cells. *Ldha* expression was mainly detected in the Sertoli cells during neonatal testis development. Its expression appeared to be high in the spermatocytes and the spermatids; however, *Stra8-cre*-mediated deletion of *Ldha* in differentiating the spermatogonia, the spermatocytes, and the spermatids had no any detectable defectsin spermatogenesis. *Ldha* deletion in the Sertoli cells causes severe problems in spermatogenesis and fertility. Although the mutant animals were not completely sterile, we assumed that this was due to the efficiency of *Amh-cre,* as similar observations were made previously [50]. These findings established *Ldha* as a key factor in maintaining the central roles of Sertoli cells in nursing spermatogenesis.

*Ldha* acts as a multifunctional factor in Sertoli cells in order to support germ cell development. Spermatogonia reside in the basal compartment of the seminiferous tubules, which is different from the luminal compartment where meiosis and spermiogenesis occur. The metabolic requirements of pro-spermatogonia and spermatogonia are therefore different from meiotic or postmeiotic germ cells [51,52]. Spermatogonium relies on its own glycolytic pathway in order to balance self-renewal and differentiation [51]. The histological analysis revealed that the spermatogonial population and the spermatocytes were not dramatically impacted by *Ldha* deletion in the Sertoli cells, indicating that these cells are not sensitive to the decreased lactate influx from the Sertoli cells. Spermatids undergo a series of dramatic morphological changes that are dependent on the lactate from the Sertoli cells. We found that the number of round spermatids decreased and that elongating spermatids were quantitatively and qualitatively disrupted. The impairments in the sperm plasma membrane and the motility affected the fertilization capacity. The low motility was due to insufficient glycolysis and ATP production in the *Ldhc*-null sperm [20,53]; however, we also found that the most obvious phenotype in the *Ldha*-cKO mice was the defects in the sperm plasma membrane and acrosome reactions, indicating that the function of lactate catalyzed by *Ldha* is not limited to energy production.

A novel finding of the present study established a functional link between lactate and choline metabolism in the testis. We demonstrated that *Ldha* ablation in the Sertoli cells changed the metabolic profile of sperm. The changes in the lactate metabolism in the Sertoli cells impacted the lipid metabolism in the sperm; notably, linoleic acids and other unsaturated fatty acids accumulated. Exogenous conjugated linoleic acid supplementation reduced the sperm quality [54]. Several metabolites that were reduced in the sperm have indispensable roles in spermatogenesis. For example, phosphocreatine is an energy source in spermatozoa and it participates in membrane fusion [55]. Among them, choline is the most important metabolite that is affected by *Ldha* deletion. Choline was found to be essential for the sperm motility and the mating behavior of fruit flies [56]. In mammals, the choline metabolism is tightly controlled in the testis, and its dysregulation is commonly associated with defects in the sperm function and infertility [45,57,58]. Choline-containing phosphatidylcholine is required in order to sustain the integrity of the plasma membrane of sperm and, thus, it plays crucial roles in acrosome reactions and capacitation [59,60]. Methionine–choline restriction significantly reduced the sperm concentration, the sperm motility, and the percentage of sperm with normal morphology [61]. Dietary supplementation with choline alleviated the fluoride-induced decline in the sperm concentration [62]. In the present study, we found that choline supplementation completely rescued the spermatogenic defects that were caused by *Ldha* deletion in the Sertoli cells. However, how the lactate metabolism in the Sertoli cells impacts the choline function in sperm is currently unclear. The functional link between these two metabolites in spermatogenesis and other cellular systems awaits further investigation.

## 5. Conclusions

In summary, we detected *Ldha* expression in spermatogenic cells and Sertoli cells. Conditional knockout experiments revealed that *Ldha* in spermatogenic cells was dispensable for spermatogenesis, but its function in the Sertoli cells was crucial for sperm function and fertility. *Ldha*-directed lactate production in the Sertoli cells regulates the motility, the membrane integrity, and the fertilization capacity of the sperm. Importantly, we identified that choline supplementation rescued the spermatogenic defects that were induced by *Ldha* loss of function. These findings highlight the key function of the Sertoli cells in maintaining spermatogenesis and establish a functional link between the lactate production and the choline metabolism in the testis. The outcomes of this study likely have therapeutic applications in other animals.

## Figures and Tables

**Figure 1 biology-11-01791-f001:**
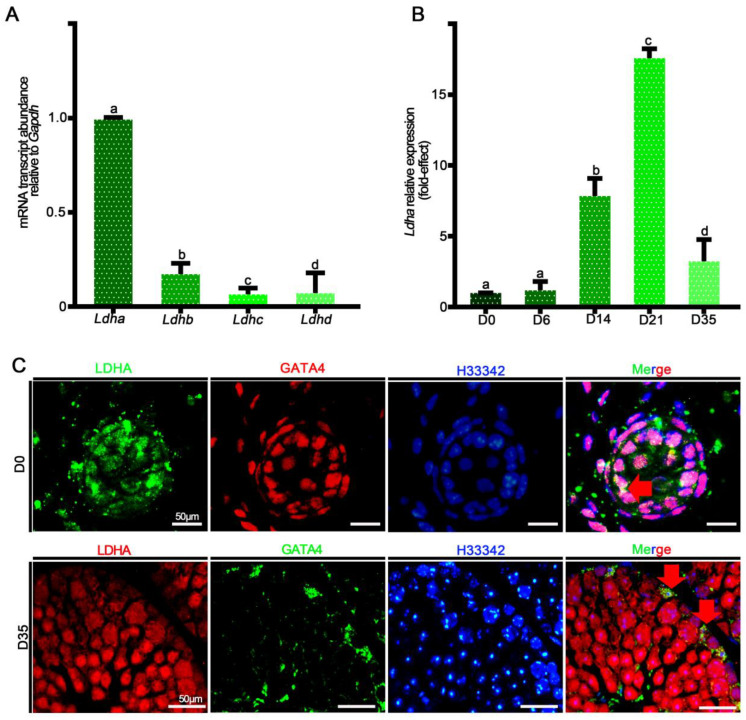
The relative mRNA expression of *Ldha-d* and protein localization of LDHA. (**A**) Analysis of *Ldha-d* mRNA expression in mouse testes. (**B**) Quantification of the mRNA expression of *Ldha* in *Sox9^+^* cells at different developmental stages. Data were analyzed as the mean ± s.e.m. of 3 mice per stage. a, b, c, and d denote significance at *p* < 0.05. (**C**) Co-immunofluorescence staining for LDHA and GATA binding protein 4 (GATA4) at D0 and D35 in cross-sections of testes. (Scale bar = 50 μm). The arrow indicates overlapping staining.

**Figure 2 biology-11-01791-f002:**
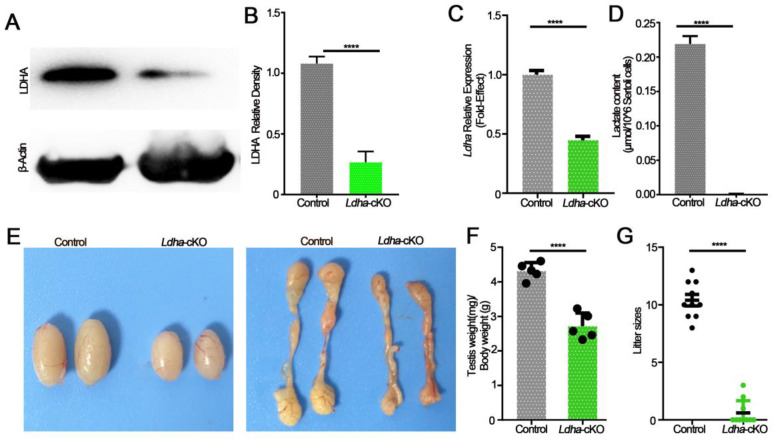
LDHA deficiency in Sertoli cells leads to smaller testes and subfertility. (**A**) Western blot image showing LDHA expression in the control and *Ldha*-cKO testes. (**B**) Quantitative data of LDHA in three-month-old male control mice and *Ldha*-cKO mice. (n = 5). (**C**) An analysis of *Ldha* mRNA expression in the control and *Ldha*-cKO testes of three-month-old males. (**D**) A comparison of LHDA content in three-month-old male control mice and *Ldha*-cKO mice. (**E**) Representative images of testes and epididymis from three-month-old male control and *Ldha*-cKO mice. (**F**) Ratios of testes-to-body of the control and *Ldha*-cKO mice (n = 5). (**G**) Comparisons of litter size between the control and *Ldha*-cKO mice (n = 5). All quantitative data are presented as the mean ± s.e.m. for n = 5 independent mouse replicates. **** indicates a significant difference of *p* < 0.0001. Gray indicated control, and green indicated *Ldha*-cKO mice.

**Figure 3 biology-11-01791-f003:**
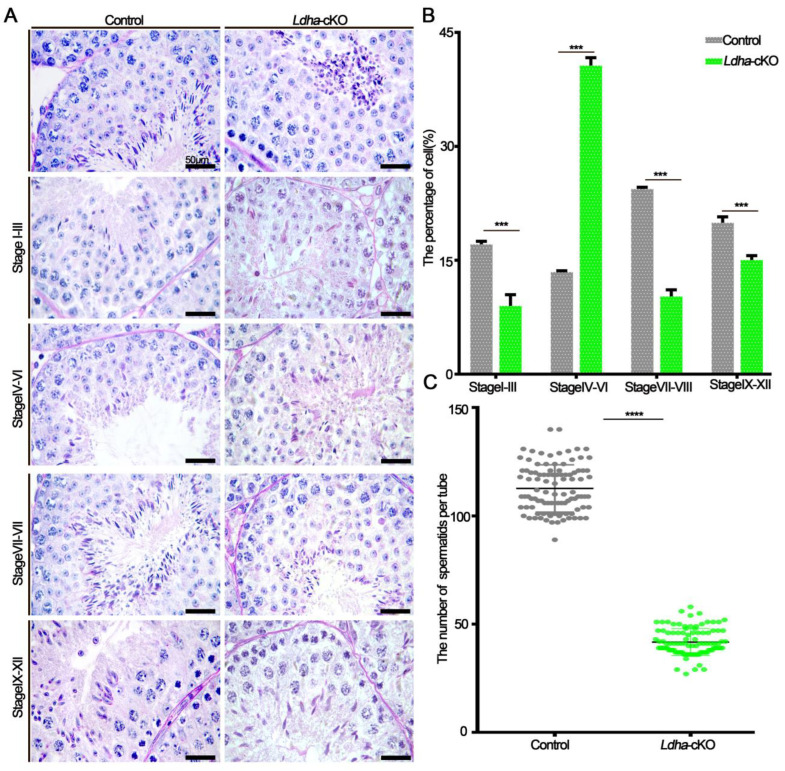
*Ldha* deletion in Sertoli cells altered the spermatogenesis stage. (**A**) Representative images of periodic acid–Schiff (PAS)-stained testes from the three-month-old control and *Ldha*-cKO mice (scale bar = 50 μm). (**B**) Distribution of stages in the PAS-stained tests from the male control and *Ldha*-cKO mice at the age of three months (scale bar = 50 μm). At least 1000 tubes were counted per group. *** indicates a significant difference of *p* < 0.001. (**C**) Round spermatids were counted in stages VII–VIII of Seminiferous tubules in the male control and *Ldha*-cKO mouse testes. (n = 5). At least 200 tubes of stages VII–VIII of Seminiferous tubules were counted per group. **** indicates a significant difference of *p* < 0.0001.

**Figure 4 biology-11-01791-f004:**
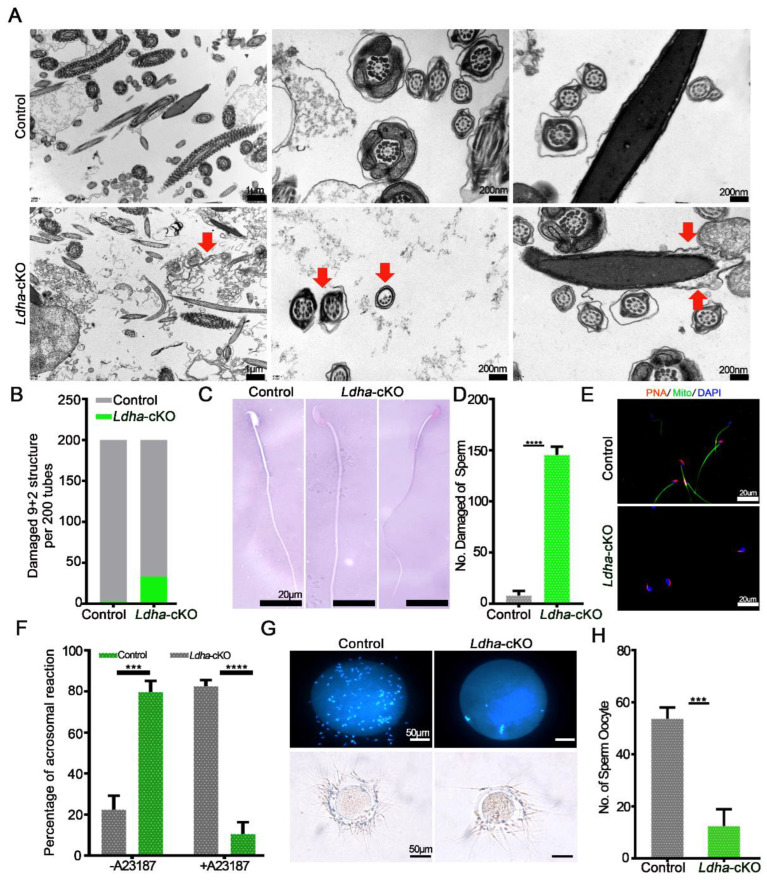
*Ldha* deletion in Sertoli cells changed the sperm structure. (**A**) Representative TEM images of sperm from the control and *Ldha*-cKO male mice. The red arrow indicates increased cellular debris in the damaged sperm plasma membrane and a damaged “9 + 2” structure. (Scale bars = 200 nm). (**B**) Rate of damage to the “9 + 2” structure of the sperm per 200 tubes in the control and *Ldha*-cKO male mice. (**C**) Representative images and the number (**D**) of eosin-aniline-black-stained sperm obtained from the male control mice and male *Ldha*-cKO mice at three months of age (scale bar = 20 μm). **** indicates a significant difference of *p* < 0.0001. n > 3. (**E**) Acrosome reactions and rate (**F**) induced by A23187 from the control and *Ldha*-cKO male mice. (Scale bars = 50 μm). At least 1000 sperm were counted for each genotype. *** indicates a significant difference of *p* < 0.001. **** indicates a significant difference of *p* < 0.0001. (G) Representative images and the number (**H**) of sperm binding to ZP from the control and *Ldha*-cKO male mice. (Scale bar = 50 μm). At least 50 oocytes were counted for each genotype. *** indicates a significant difference of *p* < 0.001.

**Figure 5 biology-11-01791-f005:**
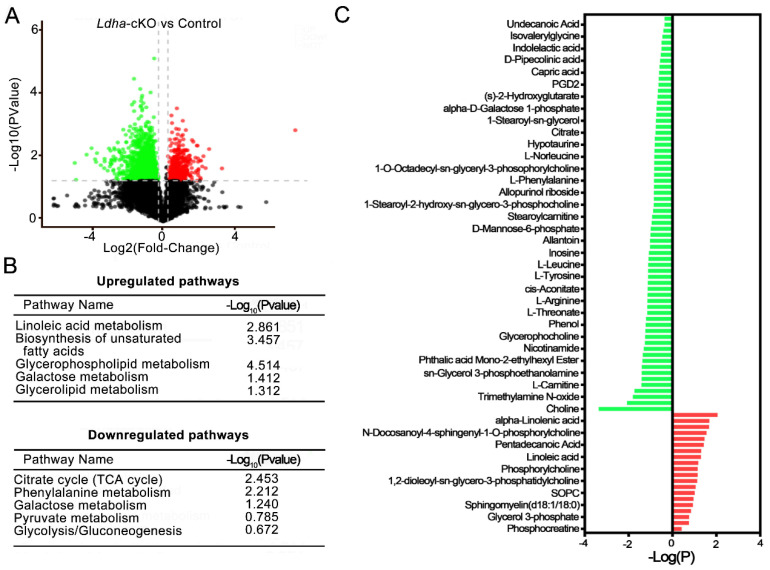
Metabolomic dysregulations in sperm from *Ldha*-cKO compared to control male mice. (**A**) The volcano plot displays the upregulated (red dots) and downregulated (green dots) metabolites. (**B**) Overview of up—and downregulated pathways in *Ldha*-cKO mouse sperm. (**C**) Overview of up- and downregulated metabolites of sperm in the male control (n = 6) and *Ldha*-cKO mice (n = 7).

**Figure 6 biology-11-01791-f006:**
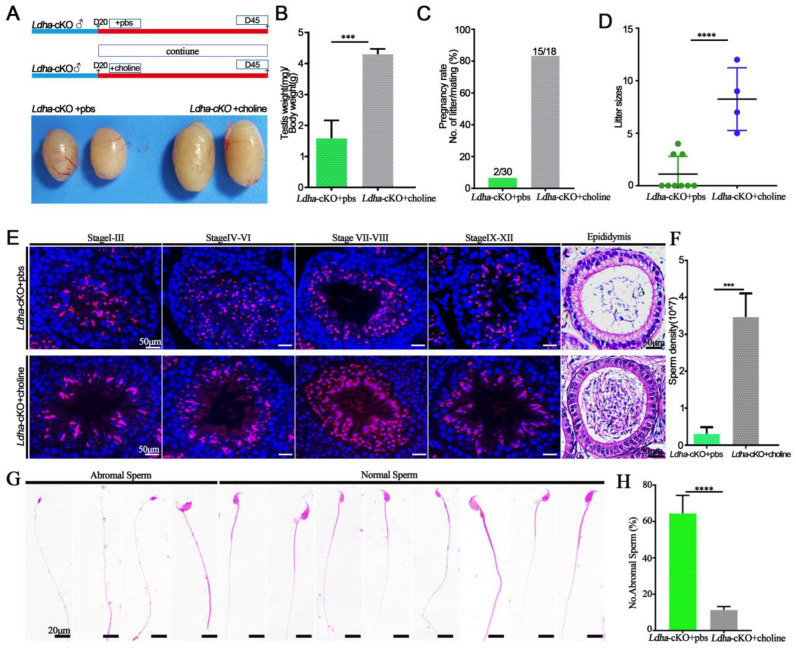
Choline can rescue the phenotype of *Ldha*-cKO mice. (**A**) Flowchart of the rescue experiment sets. (**B**) Ratios of testes-to-bodyweight of the *Ldha*-cKO mice treated with PBS as a control (n = 3) and *Ldha*-cKO (n = 5) mice treated with choline at three monthsold. *** indicates a significant difference of *p* < 0.001. (**C**) Pregnancy rate was calculated as the ratio of the number of females pregnant to the number of females with successful mating from the *Ldha*-cKO mice treated with PBS as a control (n = 3) and *Ldha*-cKO (n = 5) mice treated with choline. (**D**) Comparisons of litter sizes from the *Ldha*-cKO mice treated with PBS as a control (n = 3) and *Ldha*-cKO (n = 5) mice treated with choline at three monthsold. **** indicates a significant difference of *p* < 0.0001. (**E**) Representative images of PNA of testes and H&E staining of cauda epididymis from the *Ldha*-cKO mice with added PBS as a control (n = 3) and *Ldha*-cKO (n = 5) mice with added choline at three months old (scale bars = 50 μm). (**F**) Comparisons of sperm concentration from the *Ldha*-cKO mice treated with PBS as a control (n = 3) and *Ldha*-cKO (n = 5) mice treated with choline for 25 days. *** indicates a significant difference of *p* < 0.001. (**G**) Image of abnormal and normal sperm. (Scale bar = 20 μm). (**H**) Comparisons of abnormal sperm from the *Ldha*-cKO mice treated with PBS as a control (n = 3) and *Ldha*-cKO (n = 5) mice treated with choline. In a mouse, 200 sperm were counted and used to calculate the total percentage of abnormal sperm morphology. **** indicates a significant difference of *p* < 0.0001.

## Data Availability

All data are provided in the Appendix A.

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
