# Peer review of "Ldha-Dependent Metabolic Programs in Sertoli Cells Regulate Spermiogenesis in Mouse Testis"

_biology, 2022, doi:10.3390/biology11121791_

Round 1

Reviewer 1 Report

The authors investigated the function of LDHA on testis and sperm by a number of analytical means. Though the volume of data is abundant, sample size is small in some experiments (n=3). Additionally, the manuscript was not prepared in the proper manner and had not been carefully checked as seen in attached file

Author Response

Response to Reviewer #3 Comments

The authors investigated the function of LDHA on testis and sperm by a number of analytical means. Though the volume of data is abundant, sample size is small in some experiments (n=3) Additionally, the manuscript was not prepared in the proper manner and had not been carefully checked as seen in attached file

Response: We have sacrificed more animals and added the new data in the revised manuscript. We also have changed manuscription for proper manner according to your detailed suggestions.

1 In line 116, For the fertility test, adult control and Ldha-cKO males (n=3 for each genotype) were paired with wild-type females. Each male (2 months old) was paired with 4 females for 3 months, and litter sizes were quantified. The reviewer thought: I wonder if 3 samples are enough.

Response: Because this line is kept on the same genetic background and the results was consistent, for fertility study, we used 3 mice. In the metabonomics analysis, 6 samples were used for each genotype.

2 In line 150, Epididymal sperm were collected in 300 μl human tubal fluid (HTF) and were processed for PNA and mitochondrial staining as described previously [35]. Brief explanation is required. And PNA Explanation for this abbreviation emerges later (line 209)

Response: We agree with the reviewer and revised the manuscript accordingly. The testes were fixed in 4% paraformaldehyde solution (PFA), and the slides were boiled in 10 mM sodium citrate buffer for antigen retrieval. Then the PNA(1:500, L32458, Invitrogen™) with red fluorescence was incubation at 4℃ for overnight. Then the DAPI was added following a microscope. PNA was explanation as peanut agglutinin.

3 In line 163, To investigate changes in sperm metabolism caused by Ldha loss of function, sperm samples (n=6 for control and n=7 for Ldha-cKO animals) were collected and lysed in 80% cold methanol. RExact temperature must be described in such scientific report.

Response: Revised accordingly. To investigate changes in sperm metabolism caused by Ldha loss of function, sperm samples (n=6 for control and n=7 for Ldha-cKO animals) were collected and lysed in 80% cold methanol which maintain at 4℃.

4 In line 191, while the nontreated Ldha-cKO groups received drinking water supplemented with PBS. Reviewer3: PBS is a buffer solution, and not something to be supplemented.

Response: We have revised the text accordingly.

5 In line 194, For the fertility test, adult added choline and added pbs males (n=3 for Ldha-cKO added pbs and 194 n=5 for Ldha-cKO choline treatment) were paired with wild-type females. Reviewer3: Must be capitalized.

Response: We have changed capitalize PBS.

6 In line 202, Briefly, cumulus-free oocytes were placed in a drop of HTF medium and inseminated with capacitated sperm (2×105 sperm/ml). Reviewer3: Explanation for abbreviation should be done when emerges first.

Response: Revised accordingly.

7 In line 207, the mature spermatozoa isolated from the cauda epididymis were incubated in Human Tubal Fluid (HTF). Same as mentioned above.

Response: Revised accordingly.

8 In line 209, The capacitated spermatozoa were treated with calcium ionophore A23187 (C9400-5MG; Sigma, Germany) to induce the acrosomal reaction (AR). Same as mentioned above.

Response: Revised accordingly. The capacitated spermatozoa were treated with calcium ionophore A23187 (C9400-5MG; Sigma, Germany) to induce the AR.

9 In line 211, The acrosome was stained with fluorescein-conjugated peanut agglutinin (PNA) lectin labeling (50 mg/mL). Same as mentioned above.

Response: Revised accordingly. The acrosome was stained with fluorescein-conjugated PNA lectin labeling (50 mg/mL)

10 In line 215, In vitro fertilization and embryo culture. I wonder how the authors use embryo, oocyte and egg interchangeably.

Response: An embryo refers to the early developmental stage following the fertilization of an egg by sperm containing 2-cell and blastosphere. Oocyte refers to an oocyte is an immature egg (an immature ovum). Oocytes develop to maturity from within a follicle. We have replaced egg with oocyte to ovoid confusion.

11 In line 226, 200 sperms were used to analyze process of anomaly and acrosomal reaction detection. These explanations should be located in each sub-chapter like "2.8. Sperm membrane staining and acrosomal reaction analysis".

Response: Revised accordingly.

12 In line 229, Differences between means were examined using the t test function of GraphPad Prism 7 (La Jolla, CA, USA). T test cannot be used for comparison among more than 3 groups.

Response: T-test was used to compare the differences between the two groups One-way ANOVA was used to compare the difference when more than 2 groups.

13 In line 237, we isolated Sox9+ cells from adult testes. Reviewer3: Please state whether the testes were originated form KO or control mice.

Response: We isolated Sox9+ cells from wild type adult testes

14 In line 251, (Supplemental Figure 1) Supplement figures are not provided to reviewer.

Response: We have provided Supplemental Figure1-11 in the revised manuscript.

15 In line 263, three-month-old male control and Ldha-cKO mice. Reviewer3: There are no indications of which is which.

Response: We had labeled indications in the Figure2

16 In line 273, We asked whether Ldha in Sertoli cells was crucial for spermatogenesis and fertility. Reviewer3: I feel this expression is strange.

Response: We detected whether Ldha in Sertoli cells was crucial for spermatogenesis and fertility.

17 In line 278, Ldha expression was greatly reduced in the testes of Ldha-cKO animals. I wonder why Ldha express even in KO mice.

Response: Ldha is not only expressed in Sertoli cells, so in the RNA isolated from testis, the ldha expression can be detected in germ cells.

18 In line 279, As expected, the content of lactic acid produced by Sertoli cells was decreased by more than 90% in Ldha-cKO testes compared to that in control testes (Figure 2C). Reviewer3: Content of LDHA not lactic acid is shown in Fig. 2C.

Response: We had changed in Figure2C as lactate content

19 In line 282, The testis weight of Ldha-cKO animals was reduced by 44.26% compared to controls, and the epididymis weight of the mutants was also significantly reduced by 24.21% at 3 months of age. Mutant is different from KO.

Response: We had changed as: The testis weight of Ldha-cKO animals was reduced by 44.26% compared to controls, and the epididymis weight of the Ldha-cKO animals was also significantly reduced by 24.21% at 3 months of age.

20 In line 285, because only 3 of the 15 Ldha-cKO mice sired pups. Reviewer3: Please provide pregnancy rates of control and KO mice.

Response: We had changed as: because only 3 of the 15 Ldha-cKO mice sired pups (10.41±0.49 vs 0.60±0.34)

21 In line 287, showed that sperm concentration was reduced by 47.16% (3.6x105/mL vs 6.7x107/mL control) in Ldha-cKO mice compared to controls (Supplemental Figure 2A&C). Reviewer3: Please provide S.E.M for each data.

Response: We had changed as: showed that sperm concentration was reduced by 47.16% compared to controls (3.6x105±0.70/mL vs 6.7x107±8.80/mL control) in Ldha-cKO mice compared to controls (Supplemental Figure 2A&C).

22 In line 291, As a result, the percentages of grade A and B sperm. There is no explanation for sperm grading.

Response: We had changed as: Sperm was classified into four grades A, B, C, and D according to motility, where grade A indicated rapid forward movement, grade B slow forward movement, grade C in situ movement, and grade D immobility.

23 In line 293, Additionally, apoptosis in the testis was increased 132.68% in the Ldha-cko mice compared to the controls Reviewer3: I can’t find any explanation for apoptosis experiment in Materials and methods.

Response: In the materials and methods 2.2 The slides were then incubated with the appropriate secondary antibody for 2 h. Ter-minal deoxynucleotidyl transferase (TdT) mediated dUTP-biotin nicked end-labeling (TUNEL), which always used to detect apoptosis assay was performed by using In Situ Cell Death Detection kit (TUNEL, Beyotime, Shanghai, China) to detect DNA frag-mentation in apoptotic cells.

24 Reviewer3: In Figure3 Explanation for (C) is missing.

Response: (C) Comparsion of stages in the PAS-stained tests from male control and Ldha-cKO mice at age 3 months (n = 5). *** indicates a significant difference of P < 0.001.

25 In line 315, To investigate the cause of significantly declined fertility in Ldha-cKO males, we performed histological analysis on PAS-stained testicular cross-sections of adult testes. Reviewer3: This explanation is a duplicate of line 314-315

Response: We had changed as: To investigate the cause of significantly declined fertility in Ldha-cKO males, we performed histological analysis on adult testes.

26 In line 317, The results showed that the percentage of seminiferous tubules at stages I-III, VII-VIII and IX-XII was significantly decreased. Reviewer3: I don't understand the meaning of hese stages, because there are no explanation for them.

Response: We had added as: Spermatogenesis can be divided into 16 steps in mouse also define the seminiferous tubules into stageâ… -â…«, along with cellular association between the developing germ cell and sertoli cell(Loeschcke, 2013).

27 Reviewer3: In Figure4F: This should be some technical mistake.

Response: Revised accordingly.

28 In line 328, Reviewer3: In Figure4H There is no explanation for ***.

Response: We had added as: *** indicates a significant difference of P < 0.001. n=3.

29 In line 334, TEM analysis of sperm revealed ultrastructural abnormalities caused by Ldha loss of function in Sertoli cells. Reviewer3: There was no explanation.

Response: We had changed as: Transmission electron microscope (TEM) analysis of sperm revealed ultrastructural abnormalities caused by Ldha loss of function in Sertoli cells.

30 In line 340, Due to the abovementioned changes, acrosome reactions induced by A23187 were completely different in sperm from control and Ldha-cKO animals. Reviewer3: Need a space between e and m.

Response: Due to the above mentioned changes, acrosome reactions induced by A23187 were completely different in sperm from control and Ldha-cKO animals.

31 In line 347, Collectively, these results provide strong evidence that LDHA function in Sertoli cells is crucial for maintaining the plasma membrane integrity and fertilization ability of sperm in mice. Reviewer3: Such statements should be located in Discussion.

Response: The results should be sum up of Figure4.

32 In line 362, 365 and 366, 48 metabolites were upregulated and 433 were decreased in the sperm of Ldha-cKO animals.

Reviewer3: No uniformity in notation of upregulated and decreased.

Response: 48 metabolites were upregulated and 433 were decreased downregulated creased in the sperm of Ldha-cKO animals. Further analysis identified that the metabolism of linoleic acid, unsaturated fatty acids, glycerophospholipid, galactose and glycerolipid was upregulated, while the citrate cycle, glycolysis, phenylalanine metabolism, and pyruvate metabolism were downregulated.

33 In line 392, Figure6D Comparisons of sperm concentration from Ldha-cKO mice treated with pbs as a control (n=3) and Ldha-cKO (n=5) mice treated with choline at three months. Reviewer3: What is the real treatment duration?

Response: Comparisons of sperm concentration from Ldha-cKO mice treated with pbs as a control (n=3) and Ldha-cKO (n=5) mice treated with choline.for 25 days.

34 In line 400, Ldha-cKO animals (n=5) with dietary choline for 25 days and examined fertility and spermatogenesis. Reviewer3: This explanation contradicts the previous ones, in Materials and methods (choline was supplemented in drinking water for 4 weeks, line 190) or in Figure legend for Fig. 6D.

Response: We supplemented control (n=3) and Ldha-cKO animals (n=5) with choline in drinking water for 25 days and examined fertility and spermatogenesis.

35 In line 401-402, The results showed that choline supplementation did not affect fertility, indicating that exogenous choline at this concentration did not cause adverse effects on spermatogenesis. Reviewer3: This expression contradicts following explanation.

Response: The results showed that choline supplementation resuce fertility effectively, indicating that exogenous choline at this concentration did not cause adverse effects on spermatogenesis.

36 In 404, In sharp contrast, the testis/body weight ratio of Ldha-cKO animals receiving choline treatment increased significantly compared to the pbs-treated mutants. Reviewer3: Same as mentioned above.

Response: In sharp contrast, the testis/body weight ratio of Ldha-cKO animals receiving choline treatment increased significantly compared to the same volume PBS which used to diluent choline treated Ldha-cKO mice.

37 In 407, which was similar to that of wild-type controls, indicating a complete rescue of fertility by choline treatment. Reviewer3: How was the pregnancy rate?

Response: which was similar to that of wild-type controls, indicating a complete rescue of fertility by choline treatment (1.42±0.68 vs 8.0±0.84).

38 In line 409, concentration increased dramatically (Figure 6D&E&F). Reviewer3: I don't find Fig. 6F.

Response: The Figure6F on the top right corner.

39 In line 411, Collectively, these data provided evidence that defects in sperm morphology and function caused by Ldha deletion in Sertoli cells were rescued by choline supplementation. Reviewer3: Actual data on sperm integrity must be shown.

Response: We had added the data in Figure6G&H to support sperm integrity.

40 In 415-420, The actions of lactate in physiology and diseases have been intensively studied recently and new roles of this end production of glycolysis in energy metabolism, signaling molecule and posttranslational modification were revealed [47,48]. It has been proposed that lactate produced by Sertoli cells is the major energy source for spermatogenic cells [9,14], however, functional experiments are warranted to investigate the specific requirements of genes responsible for lactate biogenesis and metabolism in spermatogenesis. Reviewer3: These statements are duplicate of Introduction, and not necessary.

Response: We had removed the segment.

41 In 473, Among them, choline is the most important metabolite affected by ldha deletion. Reviewer3: This should be capitalized.

Response: We had changed capitalized the L. Among them, choline is the most important metabolite affected by Ldha deletion.

42 In 495, The outcomes of this study likely have therapeutic applications in treating human infertility. Reviewer3: It would be inappropriate to refer to a human therapeutic application, since it has not been revealed whether the same thing occur in human.

Response: The outcomes of this study likely have therapeutic applications in other animals.

Reviewer 2 Report

In this article, the authors demonstrate LDHA-dependent metabolic programs in Sertoli cells is essential for spermatogenesis in mouse testis. They utilized conditional KO technology to examine the role of Ldha in Sertoli cells and germ cells of testis using Stra8-cre and Amh-Cre mice. LDHA is expressed in both cells, however only Ldha deficient mice in Sertoli cell showed impaired spermatogenesis and infertility. Surprisingly, they showed Choline supplement can rescue the infertility in Ldha deficient mice in sertoli cell. These data clearly showed strong possibility of treatment of human infertility by Choline supplement.  

Minor point

As they showed in Fig1, LDHA is highly expressed in Sertoli cell and germ cells in testis. Why only Ldha cKO in sertoli cell showed infertility? If the products from Ldha activity is essential in germ cells, it seems high expressed Ldha in germ cells can rescue the defects in Sertoli cells. Do you have any thoughts? Is the amount of products (oleic acid, N6-375 methyladenosine, citrate, urail, and choline …) important or is the supplement from Sertoli cell side important?

Author Response

 We agree with the reviewer that Ldha is highly expressed in both Sertoli cells and germ cells. We generated two conditional knockout lines in this study, Stra8-cre mediated Ldha deletion in differentiating spermatogonia and spermatocytes did not cause any defects, however, Amh-cre mediated Ldha deletion in Sertoli cells causes major defects in spermatogenesis. Therefore, we conclude that this gene has important role in sperm development through a complex regulatory network.

      We found that oleic acid, choline and other metabolites are important but not sure how these metabolites are functionally connected to lactate in Sertoli cells. We will address this question in the next manuscript.

Reviewer 3 Report

In the present study, Zhang et al. provides genetic evidence that lactate production in Sertoli cells controlled by the lactate dehydrogenase A gene is crucial for murine spermatogenesis. They found that Ldha deficiency in Sertoli cells caused spermatogenic defects and adversely impacted sperm function. More importantly, they found that supplementation of Ldha-cKO animals with dietary choline rescued spermatogenic defects and fertility. This work sheds new light on the functional role of Sertoli cells in regulating spermatogenesis and identifies a candidate molecule that has potential roles in treating male infertility caused by metabolic disorders. This work provide insightful cues in understanding of the etiology of male infertility of metabolic disorder origin, but some critical concerns have to be addressed prior to consideration of acceptance.

1. The immunostaining of LDHA is not convincing. The localization of LDHA, based on the staining results in our lab, is predominantly in the cytoplasm, but it is ubiquitously expressed across the cell or in the nucleus in this study. Authors should perform staining on frozen section or with different antibodies to validate their staining result.

2. The nucleus staining should be included in images.

3. In figure 2C, the figure should be related to lactate but not LDHA content? And if possible, replace the figure 2A with immunostaining or immunoblotting of LDHA in isolated Sertoli cells to give a better view of deletion efficiency.

4. While defective spermatogenesis was observed in Ldha-cKO males, what the bona fide contribution of Ldha to spermatogenesis? Was it due to the attenuation in lactate supply to the developing germ cells or due to other defects in Sertoli cells per se caused by Ldha deletion? Could lactate supplementation rescue the defective spermatogenesis seen in Ldha-cKO mice? This is a mandatory result to support the underlying mechanism.

5. Authors performed metabolic profiling of sperm derived from control and Ldha-cKO mice and identified choline as the metabolite with the most down-regulation. Intriguingly however, choline supplementation rescued spermatogenic defects in Ldha-cKO mice. These findings are interesting but, to some extents, does not make sense. Typically in this case the metabolic profiling should be perform against testis and an improvement in defective spermatogenesis then will be well acceptable by dietary intervention. Did the author determine choline content in the testis and if so, was a decline in testicular choline content observed in Ldha-cKO mice. Therefore, as to the results in figure 6, they are somehow not quite convincing to me. I prefer to accept the findings, if any, of improvement in sperm functions by choline supplementation. Why not simply perform lactate supplementation to see if the defective spermatogenesis in Ldha-cKO mice could be rescued? Additionally, to make their data more convincing, author should explore if lactate supplementation could normalize choline content in sperms or in testis of Ldha-cKO mice.

Author Response

Response to Reviewer #2 Comments

In the present study, Zhang et al. provides genetic evidence that lactate production in Sertoli cells controlled by the lactate dehydrogenase A gene is crucial for murine spermatogenesis. They found that Ldha deficiency in Sertoli cells caused spermatogenic defects and adversely impacted sperm function. More importantly, they found that supplementation of Ldha-cKO animals with dietary choline rescued spermatogenic defects and fertility. This work sheds new light on the functional role of Sertoli cells in regulating spermatogenesis and identifies a candidate molecule that has potential roles in treating male infertility caused by metabolic disorders. This work provide insightful cues in understanding of the etiology of male infertility of metabolic disorder origin, but some critical concerns have to be addressed prior to consideration of acceptance.

  1. The immunostaining of LDHA is not convincing. The localization of LDHA, based on the staining results in our lab, is predominantly in the cytoplasm, but it is ubiquitously expressed across the cell or in the nucleus in this study. Authors should perform staining on frozen section or with different antibodies to validate their staining result.

Response: We agree with the reviewer and conducted the staining on frozen section using a different antibody from Cell Signaling Technology. We found that in the testis, Ldha is ie expressed in across the cell. We have shown the image in the revised manuscript.

  1. The nucleus staining should be included in images.

Response: We have added the nucleus staining in images.

  1. In figure 2C, the figure should be related to lactate but not LDHA content? And if possible, replace the figure 2A with immunostaining or immunoblotting of LDHA in isolated Sertoli cells to give a better view of deletion efficiency.

Response: We completely agree with the reviewer and conducted immunoblotting of Ldha. Now the revised figure clearly shows a significantly declined Ldha content in the conditional knockout mice.

  1. While defective spermatogenesis was observed in Ldha-cKO males, what the bona fidecontribution of Ldha to spermatogenesis? Was it due to the attenuation in lactate supply to the developing germ cells or due to other defects in Sertoli cells per se caused by Ldha deletion? Could lactate supplementation rescue the defective spermatogenesis seen in Ldha-cKO mice? This is a mandatory result to support the underlying mechanism.

Response: We thank the reviewer for this suggestion. We have conducted lactate supplementation experiment and found that adding lactate in the culture medium cannot rescue the defects caused by Ldha deletion. We have included the data in the revised supplemental Figure 11. From these data, we think this phenotype is caused by other defects in Sertoli cells. From metabolism analysis, we found concentrations of many choline containing compounds were changed and we tried to supplement choline and surprisingly found that the phenotype was rescued.

  1. Authors performed metabolic profiling of sperm derived from control and Ldha-cKO mice and identified choline as the metabolite with the most down-regulation. Intriguingly however, choline supplementation rescued spermatogenic defects in Ldha-cKO mice. These findings are interesting but, to some extents, does not make sense. Typically in this case the metabolic profiling should be perform against testis and an improvement in defective spermatogenesis then will be well acceptable by dietary intervention. Did the author determine choline content in the testis and if so, was a decline in testicular choline content observed in Ldha-cKO mice. Therefore, as to the results in figure 6, they are somehow not quite convincing to me. I prefer to accept the findings, if any, of improvement in sperm functions by choline supplementation. Why not simply perform lactate supplementation to see if the defective spermatogenesis in Ldha-cKO mice could be rescued? Additionally, to make their data more convincing, author should explore if lactate supplementation could normalize choline content in sperms or in testis of Ldha-cKO mice.

Response: From the metabolic profiling of sperm from control and Ldha-cKO mice, we found choline and several choline containing metabolites were significantly different. Therefore, we conducted choline supplementation experiment, but we did not expect to find that choline completely rescued spermatogenic defects caused by Ldha deletion in Sertoli cells. We thought the membrane defects may be rescued because some choline-containing phospholipids are important component of sperm membrane. We did lactate supplementation experiment as suggested by the reviewer. Now we think the clarity of this manuscript is greatly improved. We wanted to know how reduction of lactate production in Sertoli cells affected Choline and choline containing compounds.

Round 2

Reviewer 1 Report

There still remains sevral points to be revised.

Author Response

12 In line 229, Differences between means were examined using the t test function of GraphPad Prism 7 (La Jolla, CA, USA). T test cannot be used for comparison among more than 3 groups.

Response: T-test was used to compare the differences between the two groups Oneway ANOVA was used to compare the difference when more than 2 groups.

>Which statistical method did the authors use for post hoc analysis?

Response: In our study,T-test was used to compare the differences between the control and Ldha-cko mice .

21 In line 287, showed that sperm concentration was reduced by 47.16% (3.6x105/mL

vs 6.7x107/mL control) in Ldha-cKO mice compared to controls (Supplemental Figure2A&C). Reviewer3: Please provide S.E.M for each data.

Response: We had changed as: showed that sperm concentration was reduced by

47.16% compared to controls (3.6x105±0.70/mL vs 6.7x107±8.80/mL control) in LdhacKO mice compared to controls (Supplemental Figure 2A&C).

> reduced by 47.16% in Ldha-cKO mice (3.6 ± 0.70 x 105/mL) compared to controls

(6.7 ± 8.80 x 107/mL). Is ± 8.80 correct (which is bigger than the mean number)?

Response: We had re-analysed the result, the 6.7 ± 1.60 x 107/mL is correct.

22 In line 291, As a result, the percentages of grade A and B sperm. There is no

explanation for sperm grading.

Response: We had changed as: Sperm was classified into four grades A, B, C, and D according to motility, where grade A indicated rapid forward movement, grade B slow forward movement, grade C in situ movement, and grade D immobility.

> What are the specific thresholds that distinguish rapid, slow and in situ movement?

Response: In line337, Sperm were classified into four grades according to the World Health Organization, A, B, C, and D, according to motility, where grade A indicated rapid forward movement, in which the velocity should be ≥25μm/s under the microscope, and 25μm is five times the length of a sperm head, or half the length of a sperm tail. grade B indicated slow forward movement, in which the velocity ≤25μm/s, grade C indicated in situ movement or a curve movement but can be active, and grade D indicated immobility and some of which may be live and some of which may be dead.

23 In line 293, Additionally, apoptosis in the testis was increased 132.68% in the Ldhacko mice compared to the controls Reviewer3: I can’t find any explanation for
apoptosis experiment in Materials and methods.

Response:
In the materials and methods 2.2 The slides were then incubated with the
appropriate secondary antibody for 2 h. Ter-minal deoxynucleotidyl transferase (TdT)
mediated dUTP-biotin nicked end-labeling (TUNEL), which always used to detect
apoptosis assay was performed by using In Situ Cell Death Detection kit (TUNEL,
Beyotime, Shanghai, China) to detect DNA frag-mentation in apoptotic cells.

> “is” is needed instead of ”always”

Response: Revised accordingly.

24 Reviewer3: In Figure3 Explanation for (C) is missing.

Response:
(C) Comparison of stages in the PAS-stained tests from male control and
Ldha-cKO mice at age 3 months (n = 5). *** indicates a significant difference of P <
0.001.

> I can’t find this description in the legend for Fig. 3 in the manuscript.

Response: Revised accordingly.

29 In line 334, TEM analysis of sperm revealed ultrastructural abnormalities caused
by Ldha loss of function in Sertoli cells. Reviewer3: There was no explanation.
Response: We had changed as: Transmission electron microscope (TEM) analysis of
sperm revealed ultrastructural abnormalities caused by Ldha loss of function in Sertoli
cells.
12 In line 229, Differences between means were examined using the t test function of
GraphPad Prism 7 (La Jolla, CA, USA). T test cannot be used for comparison among
more than 3 groups.
Response: T-test was used to compare the differences between the two groups Oneway ANOVA was used to compare the difference when more than 2 groups.
>Which statistical method did the authors use for post hoc analysis?
>We use T-test in the

> The method for TEM is necessary to be stated

Response: In line172, Then the material was dehydrated and embedded in Epon resin; ultrathin sections were contrasted with 3% uranyl acetate and lead citrate and analyzed in A JEOL JM-1400Plus electron microscope (Tokyo, Japan) was used to capture images at 6000. 6000× and 1200012000× magnifications.

37 In 407, which was similar to that of wild-type controls, indicating a complete rescue
of fertility by choline treatment. Reviewer3: How was the pregnancy rate?

Response: which was similar to that of wild-type controls, indicating a complete
rescue of fertility by choline treatment (1.42±0.68 vs 8.0±0.84).

> ”vs 8.0±0.84” is redundant since the same information was shown just 1 line before.
“Pregnancy rate” is the percentage of pregnancies to number of mating, which is
different from the litter size. To evaluate the fertility of sperm, reproductive efficiency
including the pregnancy rate and litter size should be investigated.

Response: We take out the 8.0±0.84 as redundant. In Figure 6C, Pregnancy rate(2/30 in Ldha-cKO+pbs vs 15/18 in Ldha- cKO +choline) was added to evaluate the fertility of sperm.

38 In line 409, concentration increased dramatically (Figure 6D&E&F). Reviewer3: I
don't find Fig. 6F.

Response: The Figure6F on the top right corner.

> Figures must be arranged in alphabetical order

Response: Revised accordingly.

Line 312-315: Why the authors use different style (1.08±0.03-fold and 127%) to
mention the same meaning, which is just misleading. “reduced by 127%” seems
strange. Not “reduced by 56%”?

Response: Figure2B was indicated the LDHA protein density in control and Ldha-cKO mice. Figure2C was indicated the Ldha mRNA level, we had calculate again as reduced by 56%

Line 444, 446, 448, 464: It is not clear what these numbers indicate (what are unit for
them?). If these indicate numbers of abnormal sperms, the total numbers of observed
sperms are mandatory. Additionally, the authors explained that “the percentage of
viable spermatozoa was determined by eosin-aniline black staining” (Line 213-214).

Response: We added the unit in Figure6H. “In a mouse, 200 sperms were counted and used to calculate the total percentage of abnormal sperm morphology” was added in supplement table.

Reviewer 3 Report

The authors largely addressed my concerns about their study. The authors should check the reference section, some references are not properly listed.

Author Response

We had examined the reference.